# Human Skeletal Muscle Cells Derived from the Orbicularis Oculi Have Regenerative Capacity for Duchenne Muscular Dystrophy

**DOI:** 10.3390/ijms20143456

**Published:** 2019-07-14

**Authors:** Yukito Yamanaka, Nana Takenaka, Hidetoshi Sakurai, Morio Ueno, Shigeru Kinoshita, Chie Sotozono, Takahiko Sato

**Affiliations:** 1Department of Ophthalmology, Kyoto Prefectural University of Medicine, Kyoto 602-8566, Japan; 2Center for iPS Cell Research and Application, Kyoto University, Kyoto 606-8507, Japan; 3Department of Frontier Medical Science and Technology for Ophthalmology, Kyoto Prefectural University of Medicine, Kyoto 602-8566, Japan; 4Department of Anatomy, Fujita Health University, Toyoake 470-1192, Japan; 5AMED-CREST, AMED 1-7-1 Otemachi, Chiyoda, Tokyo 100-0004, Japan

**Keywords:** muscle stem cell, orbicularis oculi, cell transplantation, Duchenne muscular dystrophy

## Abstract

Skeletal muscle stem cells (MuSCs) have been proposed as suitable candidates for cell therapy in muscular disorders since they exhibit good capacity for myogenic regeneration. However, for better therapeutic outcomes, it is necessary to isolate human MuSCs from a suitable tissue source with high myogenic differentiation. In this context, we isolated CD56+CD82+ cells from the extra eyelid tissue of young and aged patients, and tested in vitro myogenic differentiation potential. In the current study, myogenic cells derived from extra eyelid tissue were characterized and compared with immortalized human myogenic cells. We found that myogenic cells derived from extra eyelid tissue proliferated and differentiated myofibers in vitro, and restored DYSTROPHIN or PAX7 expression after transplantation with these cells in mice with Duchenne muscular dystrophy. Thus, human myogenic cells derived from extra eyelid tissue including the orbicularis oculi might be good candidates for stem cell-based therapies for treating muscular diseases.

## 1. Introduction

Skeletal muscle tissue has its own repair and maintenance system which is based on adult skeletal muscle stem cells (MuSCs), including muscle satellite cells. Adult MuSCs are normally quiescent, but are activated upon muscle damage, either by muscular injury or under pathological conditions such as Duchenne muscular dystrophy (DMD). They proliferate, differentiate to enter the regenerative myogenic program, fuse with damaged myofibers, and also generate a novel population of quiescent MuSCs [1].

In this study, we generated and characterized human myogenic stem cells from extra eyelid tissue collected through blepharoplasty. Blepharoplasty is a form of surgical repair where the eyelid skin, orbicularis oculi muscle, and orbital fat are excised. As one ages, the eyelids gradually stretch, and thinning of the levator aponeurosis leads to ptosis of the upper eyelid [2]. Besides making one look older, severe ptosis can limit peripheral vision, especially the upper parts of the field of vision. Blepharoplasty is done for both esthetic and functional indications. After this surgery, extra eyelid tissue is normally discarded, although orbicularis oculi or levator muscles might be attached to them [3].

We assessed their potential as a source of myogenic stem cells able to improve the muscle regeneration of DMD in vivo. For this purpose, we generated high amounts of human myogenic CD56+CD82+ stem cells [4] from freshly isolated primary human ocular biopsy cultures and transplanted them into DMD/immunodeficient mice. It is known that CD56 is the marker that enables the isolation of human myogenic cells [5], and CD82 is novel marker for detecting human muscle stem cells [4,5]. We demonstrated that human CD56+CD82+ myogenic cells derived from the orbicularis oculi expressed high *PAX7* transcripts, and importantly, that selected cells exhibited an enhanced survival and an enhanced potency for generating DYSTROPHIN+ cells after transplantation in immunodeficient mice compared to immortalized human myoblasts. These data highlight a potential role of human myogenic cells from extra orbicularis oculi tissue to improve skeletal muscle healing and as a source of muscle stem cells in vivo.

## 2. Results

### 2.1. Primary Cultured Cells from Human Eyelid Tissue

Fresh tissue samples resected from human eyelids during blepharoptosis or corrective strabismus surgery in both male and female patients aged between 3 and 79 years old were used in this study. Before dissecting eyelids, human myogenic cells were confirmed by immunostaining with anti-DYSTROPHIN and LAMININ-a2 antibodies in the extra eyelid tissue after the surgical operation (Figure 1a−c), and human muscle satellite stem cells which were labelled with PAX7 were also detected on single myofibers (Figure 1d). Following confirmation of the existence of muscle stem cells in these tissues, we performed cell sorting with anti-CD56 antibody to isolate human myogenic cells directly from the tissues of patients of three different ages (all biopsies were from male patients, aged 7, 29, and 77 years old). All CD56-positive samples isolated from each age represented less than 1% of the myogenic cells for cell culture (0.17%, 0.06%, 0.11%, Figure 1e). Enzymatically dissociated cells from extra eyelid tissue were cultured in vitro for 10 days. These cells were mainly detected as fibroblasts because of staining with anti-FSP1 antibody [6], and not myogenic cells which can be detected through MYOGENIN (MYOG)-positive cells (Figure 1f).

For dissociating single human myogenic cells, we used extra eyelid tissues obtained from blepharoptosis or corrective strabismus surgery (left and middle panels in Figure 2a). Adipocytes (arrowheads, left panel in Figure 2b), blood capillaries (arrowheads, middle panel in Figure 2b), and myofibers (arrowheads, right panel in Figure 2b) were observed in the extra eyelid tissues obtained from the patients. These tissues were mechanically dissected (right panel in Figure 2a), enzymatically dissociated, and filtrated into single cells (Figure 2c). These cells from extra human eyelid biopsies were placed on Geltrex-coated dishes and cultured in DMEM containing 20% fetal bovine serum and basic FGF.

### 2.2. CD56-Positive Population from Primary Cultured Cells of Extra Eyelids

To exclude non-myogenic cells from primary cultured cell of extra eyelids (Figure 3a), we performed cell sorting again with anti-CD56 antibody to detect human myogenic cells from growing primary cultured cells of extra eyelid tissue. CD56-positive cells were detected as representing about 8% of total cultured cells (Figure 3b). These sorted cells (CD56+) morphologically resemble Hu5/KD3 (immortalized human myogenic cells [7]) in shape, as shown in the upper right panel of Figure 3c. They were cultured in differentiation medium (DM) for 7 days. The majority of the cells formed myotubes in vitro as MF20+ cells (Figure 3d), with myogenic differentiation index values of more than 95% of cells, although CD56-negatively sorted cells had little myogenic differentiation potential (Figure 3e). Moreover, the subpopulation of CD56+CD82+ double positive cells was detected as representing about half of the CD56+ cells in a primary cultured population of extra eyelid tissue (Figure 3f), and contained high PAX7 transcript levels (Figure 3g), marking muscle stem cells as reported [4,5].

### 2.3. Human Myogenic Cells Derived from Orbicularis Oculi after Subculturing

Before human myogenic CD56-positive cells reach confluency in cell culture dishes, they should be passaged when they are in the log phase. Cultured cells normally stop growing and change their qualities when they reach confluent because of contact inhibition. Novel therapies based on cell transplantation for DMD require the expansion of muscle stem cells ex vivo. Cultured muscle stem cells progress towards myogenic differentiation and have reduced regenerative potential in vivo [8]. To identify the myogenic potential of subculturing cells after several passages with human CD56+ cells obtained from young and aged patients, the positive percentage of CD56 and quantities of PAX7 transcript were analyzed. After the second passage of CD56+CD82+ double positive myogenic cells derived from young patients, CD56+CD82+ cells were still detected as 3.87% of total cultured cells; however, few CD56+ cells were obtained after the fifth passage (Figure 4a). In these cells, the ratio of PAX7 transcripts was also decreased according to passaging times (Figure 4b). CD56+CD82+ cells in primary cultured samples from aged patients were detected; however, the percentages of these double positive cells in total cultured cells were lower than in young patients (Figure 3f and Figure 4c). Moreover, the percentage of CD56+CD82+ double positive myogenic cells from aged patients after the second passage was as low as in fifth passaged cells from young patients (right panel in Figure 4c), and in hypoxic conditions of cell culture, the numbers of grown cells was increased. However, the percentage of CD56+CD82+ cells was decreased in this condition (Appendix A) [9]. The proportion of PAX7 transcript was also checked in passaged cells from young and aged patients. We found that higher transcripts of PAX7 were detected in CD56+CD82+ cells derived from young patients than from aged patients, and in early-passaged cells from both groups of patients (Figure 4b,d). The declining expression level of PAX7 transcripts might be due to the attenuation of PAX7-positive cells of the CD56+CD82+ population in cells of passaged and aged samples.

### 2.4. CD56CD82-Double Positive Cells from Extra Eyelids after Passaging

CD56+ cells of human eyelid origin have myogenic differentiation potential as immortalized myoblasts. To identify the myogenic potential of human muscle stem origin in vivo, tibialis anterior muscle sections from DMD mice injected with either CD56+ or CD56− cells which were grown from primary bulk population of extra eyelids in the condition of 20% oxygen for 7 days, or immortalized human myoblasts Hu5 cells as a positive transplantation control (Figure 5a) were used. Transplanted tibialis anterior (TA) muscles were stained with antibodies against human lamin A/C (hLMNA) and DYSTROPHIN (Figure 5b). This showed that the percentage of hLMNA+ (red in Figure 5b) and DYS+ cells (green in Figure 5b) in representative muscle sections was significantly increased after CD56-positive cell injection compared to CD56-negative cell injection (Figure 5c,d). We further observed that not injected human myoblasts but human CD56+ cells expressed PAX7 at 14 days after cell injection (Figure 5e). In vivo localization of these cells was also investigated using an antibody directed against LAMA2, a basal lamina protein that cover the niche of muscle satellite cells. Double-positive cells for PAX7 and hLMNA residing within the interstitial areas were unexpectedly observed in transplanted DMD mice 2 weeks after intramuscular injection, suggesting that these human cells adopt an atypical stem cell position (red arrowhead in Figure 5e). Since muscle satellite stem cells were originally defined by their location of residence [10], this result indicates that observed cells did not move from their niches, directly engrafted to the interstitial areas in regenerating state [11], or could contribute to muscle satellite cell compartment after a long period because the transplanted cells were observed at early stages of regeneration.

## 3. Discussion

Our study and previous reports also show that growing myogenic stem cells are present in the muscle tissues of human orbicularis oculi derived from extra eyelid tissue, and could be successfully isolated and cultured without compromising their myogenic differentiation ability (Figure 2 and Figure 3) [12,13]. By merely plating dissociated eyelid tissue after enzymatic treatment, not only spindle shaped cells but fibroblastic cells formed adherent colonies. Primary myogenic cells were selected by the cell surface expression of CD56 and CD82, along with spindle shaped morphology, and myogenic differentiation capacity suggests that primary cultured CD56+CD82+ double-positive cells from the orbicularis oculi of young patients resemble immortalized human myoblast Hu5 or primary human muscle cells derived from other tissues (Figure 4) [5,7]. Nevertheless, myogenic cells that we directly isolated from extra eyelids by cell sorting were rather poorly detected (Figure 1).

We investigated the characteristics of muscle stem cells from the orbicularis oculi, a group of facial muscles selectively spared or involved in different muscular dystrophy disorders. As far as the expression of proteins involved in excitation–contraction coupling is concerned, it has been reported that orbicularis oculi muscles are closer to quadriceps than to extra ocular muscles, and that several results concerning the expression of utrophin are interesting and most likely explain why in DMD patients ocular and facial muscles are spared [14,15]. Utrophin and dystrophin share a structural similarity, and utrophin can associate with dystrophin complex, serving as a link between muscle actin and the extracellular matrix [16,17]. In DMD mice, it is suggested that utrophin can compensate for the lack of dystrophin, supporting the current strategies aimed at modulating utrophin expression in the therapy for DMD [18]. In that sense, the use of facial muscles including extra eyelid tissue might be useful for the study of in vitro myogenic potential and the treatment of DMD by not only cell transplantation but the replacement of utrophin.

Taken together, we have identified the signature of muscle stem cells from orbicular oculi with myogenic differentiation capacity in vitro which make them suitable candidates for use in treatment of DMD. These myogenic stem cells from orbicular oculi were not gained at a massive scale from other muscle tissues; however, we do not need to perform unnecessary surgical procedures and can plan to obtain myogenic cells from the orbicular oculi. Obtaining a large quantity of muscle stem cells in culture would be next challenge. In conclusion, our findings are potentially important for considering non-invasive human primary tissues to retrieve myogenic cells for cell transplantation in DMD efficiently and safely.

## 4. Materials and Methods

### 4.1. Human Biopsies of Extra Eyelid Tissue

Human biopsies of extra eyelid tissue including skeletal muscle tissues were collected during ophthalmologic surgery of healthy patients at University Hospital Kyoto Prefectural University of Medicine. All methods relating the human study were performed in accordance with the guidelines and regulations of the Kyoto Prefectural University of Medicine (Permission numbers: ERB-E-74, ERB-C-491, 5/Jan/2016).

### 4.2. Primary Human Muscle Cell Culture

Human biopsies of extra eyelid tissue were minced and subjected to enzymatic dissociation with 0.1% Collagenase Type2 (Worthington, Lakewood, NJ, USA) in DMEM (WAKO, Osaka, Japan) at 37 °C for 60 min. Dissociated cells from biopsies or sorted cells were plated in DMEM containing 20% FBS and 5 ng/mL of basic FGF (WAKO), coated with Geltrex (GIBCO, Waltham, MA, USA). Fresh media was added regularly until colonies with spindle shaped cells were obtained. For cell sorting, cultured cells were detached with Accutase (Nacalai, Kyoto, Japan) from cell culture dishes, resuspended with 1% bovine serum albumin (Sigma-Aldrich, St.Louis, MO, USA) in Hank’s Balanced Salt Solution buffer (WAKO), and incubated with the monoclonal anti-human antibodies anti CD56-PE and anti-CD82-Alexa647 (BioLegend, San Diego, CA, USA). After 30 of min incubation at 4 °C, human myoblasts including muscle stem cells, defined as CD56+CD82+, were sorted by flow cytometry using a FACS JAZZ (BD, Frankli Lakes, NJ, USA). Isotype control antibodies were PE- and Alexa647-conjugated mouse IgG1 (BioLegend), filtrated with a cell strainer (BD). Cell suspensions were stained with 1 mg/mL propidium iodide (Nacalai) to exclude dead cells.

### 4.3. RT-qPCR Analyses

Total RNAs from sorted or cultured cells were extracted using Rneasy Micro Kit (Qiagen, Hilden, Germany). For quantitative PCR analyses, synthesized cDNA was prepared using the SuperScript III kit (Invitrogen) with random hexamers from mRNAs. All RT–qPCR reactions were carried out in triplicate using THUNDERBIRD SYBR qPCR Mix (TOYOBO, Osaka, Japan), and normalized to mRNA expression level of ribosomal protein L13A (RPL13a) as a control. Primer sequences (5′ to 3′) are listed as follows: RPL13a-Forward: 5′-ccctggaggagaagaggaaa-3′, RPL13a-Reverse: 5′-acgttcttctcggcctgttt-3′, PAX7-Forward: 5′-gggattccctttggaagtgt-3′, PAX7-Reverse: 5′-cggcaaagaatcttggagac-3′.

### 4.4. Immunofluorescence

Cells and skeletal muscle tissues were fixed with 4% paraformaldehyde in PBS for 10 min at 4 °C, before embedding in Frozen Section Compound (Leica Microsystems) for cryosections. Fixed samples were incubated with anti-PAX7 (DSHB; diluted 1/100), anti-MYOG (DAKO; diluted 1/100), anti-DYSTROPHIN (DAKO; diluted 1/100), anti-MyHC (MF20, R&D; diluted 1/ 200), anti-LAMA2 (Enzo Life Sciences, Farmingdale, NY, USA; diluted 1/500), anti-hLMNA (Abcam; diluted 1/200), and anti-FSP1 (Abcam; diluted 1/200) antibodies in 5% of BlockingOne (Nacalai) for overnight at 4 °C. After three washes with 0.1% of Tween20 in PBS, cells were incubated with Alexa488, Alexa594, or Alexa647-conjugated secondary antibodies (Molecular Probes; diluted 1/500). Cells were washed and mounted in *SlowFade* Diamond Antifade Mountant with DAPI (Molecular Probes, Eugene, OR, USA). Images were collected and processed to change original fluorescent colors when appropriate on the software of BZX-710 (Keyence, Osaka, Japan). For quantitation of cultured cells, numbers of dishes were analyzed from triplicate experiments.

### 4.5. Grafting into Dystrophin-Deficient Muscle Tissues

Twelve-week or older *Dmd*^-/y^ and NSG host male mice were used for engraftment of cultured cells (1.0 × 10^4^ cells per 20 microliters of PBS) into tibialis anterior (TA) myofibers [8]. Mice were anesthetized with diethyl ether before engraftment. TA muscle was removed 2 weeks after transplantation, fixed, and stained as above. For quantification, serial transverse sections were cut across the entire TA muscle, generating about 20 slides per muscle where each slide contained about 20 serial sections. Five different slides were immunostained, covering regions where most engrafted cells were present. At least four transplanted mice were analyzed per each experiment.

### 4.6. Statistics

We report the statistical data, including results of at least three biological replicates. Statistical analyses were performed with StatPlus software (AnalystSoft, mac LE, Walnut, CA, USA) to determine significant differences from a two-tailed distribution using paired or unpaired Student’s *t*-test or Dunnett’s multiple-comparisons test. *p*-values indicated on each figure are <0.01 (*). All error bars are indicated as means ± SEM (*n* = 3).

## Figures and Tables

**Figure 1 ijms-20-03456-f001:**
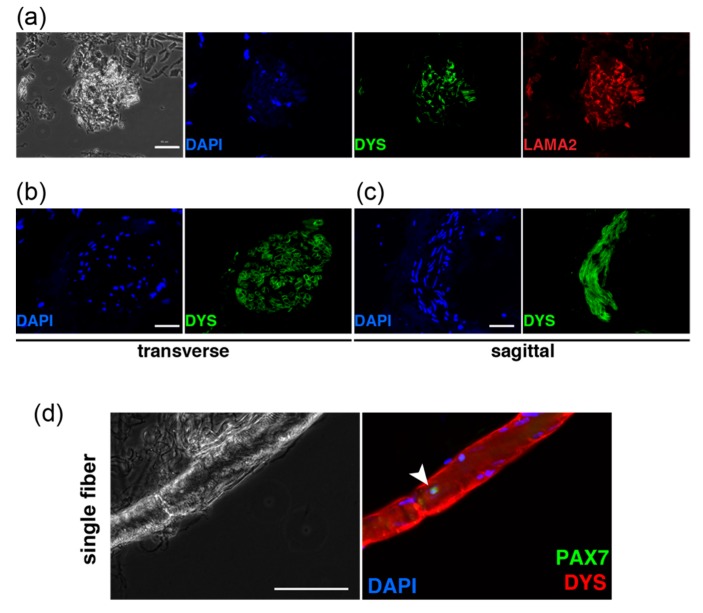
Characteristics of surgically obtained eyelid tissues and cells. (**a**) Isolated eyelid tissues were immunostained with anti-DYSTROPHIN (DYS, green) and laminin α-2 (LAMA2, red) antibodies. Scale bar, 50 μm. (**b**) Transverse section of isolated tissues containing myofibers, stained with anti-DYS (green). Scale bar, 50 μm. (**c**) Sagittal section of (**b**). Scale bar, 50 μm. (**d**) Single myofibers from extra eyelid tissues were immunostained with anti-PAX7 (arrowhead, green in right panel), and DMD (red). Scale bar, 100 μm. (**e**) FACS profiles for detecting CD56-positive cells from digested eyelid tissues. Samples from M7 (a 7-year-old male patient), M29 (a 29-year-old male patient), and M77 (a 77-year-old male patient) were analyzed. (**f**) Morphology of cultured cells from digested eyelid tissues over 10 days (left panel). Expanded cells were immunostained with anti-MYOGENIN (MYOG, green) and anti-FSP1 (red). Scale bar, 100 μm. All nuclei were stained with 4’6-diamidino-2-phenylindole (DAPI, blue).

**Figure 2 ijms-20-03456-f002:**
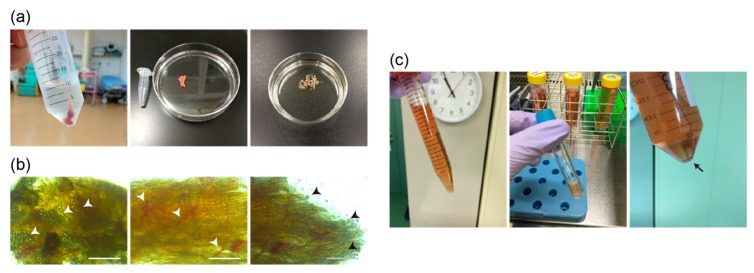
The schematic representation of collecting human skeletal muscle cells obtained from extra tissues containing orbicularis oculi muscles at the time of ophthalmic surgery. (**a**) Surgically excised eyelid tissues soaked in cold PBS solution (left panel), an example of the actual size of the extra eyelid tissue compared with a 1.5-mL microtube (middle panel), and the obtained tissue finely chopped by scissors (right panel). (**b**) Morphological features of isolated tissues, mass of lipids (arrowheads in left panel), blood capillaries (arrowheads in middle panel), and disconnected skeletal muscle fibers (arrowheads in right panel). Scale bars, 200 μm. (**c**) Chopped samples were enzymatically treated with collagenase type 2 (left panel), then filtrated after enzymatic digestion (middle panel), and centrifuged to collect single myogenic cells (arrow, right panel).

**Figure 3 ijms-20-03456-f003:**
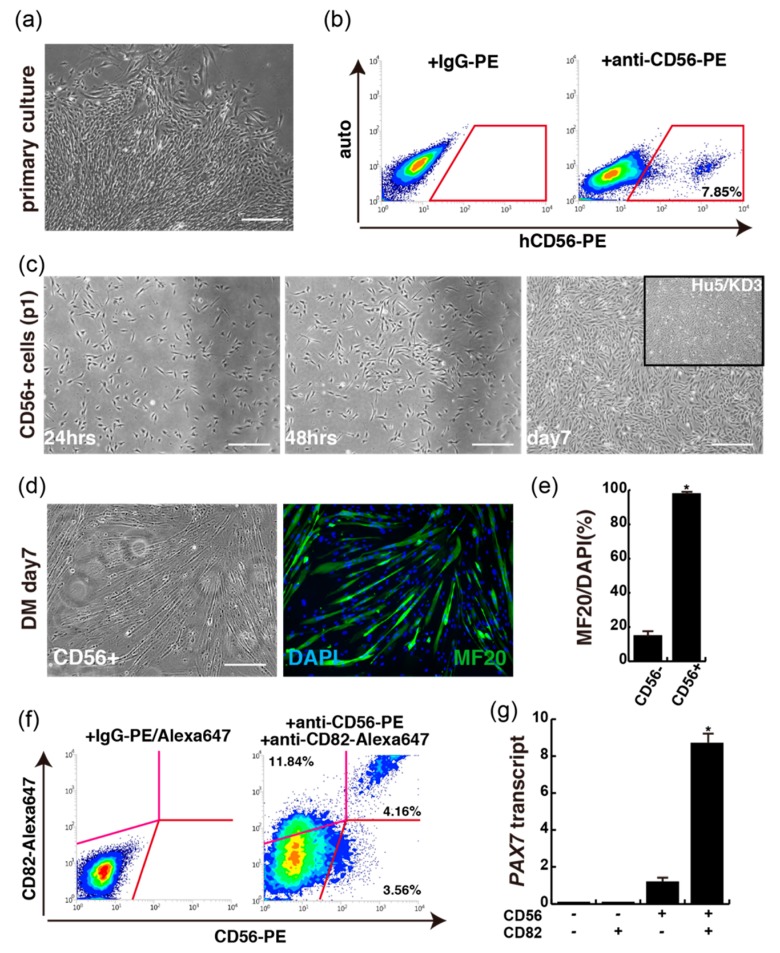
CD56+CD82+ double-positive myogenic cells from primary cultured orbicularis oculi muscles. (**a**) Image of primary cultured cells digested from isolated eyelid tissue. (**b**) FACS profile for detecting CD56-positive cells from primary cultured cells with extra eyelid tissue. (**c**) Morphological features of CD56-sorted cells (passage 1, p1) of orbicularis oculi muscles, cultured for 24 h (24hrs), 48 h (48hrs), and 7 days (day7). (**d**) Myogenic differentiation with CD56+ cells in 2% horse serum medium for 7 days (DM, day7). MF20, green; DAPI, blue. (**e**) The percentage of differentiated myogenic cells is presented as MF20/DAPI ratio. *n* = 3 independent replicates; *p*-values were determined by *t*-test from a two-tailed distribution. * *p* < 0.01. (**f**) FACS profile of CD56- and CD82- double positive cells from primary cultured orbicularis oculi muscles from extra eyelid tissue. (**g**) PAX7 transcriptome analyses of CD56+CD82+ sorted cells shown in (**f**). *n* = 3 independent replicates; *p*-values are determined by Dunnett’s multiple-comparisons test. * *p* < 0.01. All scale bars, 100 μm.

**Figure 4 ijms-20-03456-f004:**
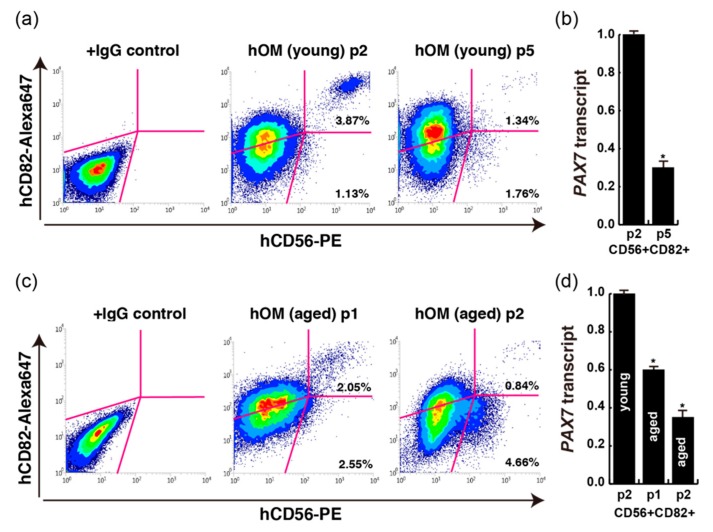
FACS analyses with each primary cultured human orbicularis oculi muscle (hOM). (**a**) FACS data of CD56+CD82+ double positive cells from cultured double positive cells (passage 2, p2; passage 5, p5) from dissected extra eyelid tissue (14-year-old male patient; young). (**b**) PAX7 transcription of CD56+CD82+ sorted cells from different passage times. *n* = 3 independent replicates; *p*-values were determined by *t*-test from a two-tailed distribution. * *p* <0.01. (**c**) FACS data of CD56+CD82+ double positive cells from cultured double positive cells (passage 1, p1; passage 2, p2) from dissected extra eyelid tissue (74-year-old male patient; aged). (**d**) PAX7 transcription of CD56+CD82+ sorted cells from different age and passage samples. *n* = 3 independent replicates; *p*-values were determined by *t*-test from a two-tailed distribution. * *p* <0.01.

**Figure 5 ijms-20-03456-f005:**
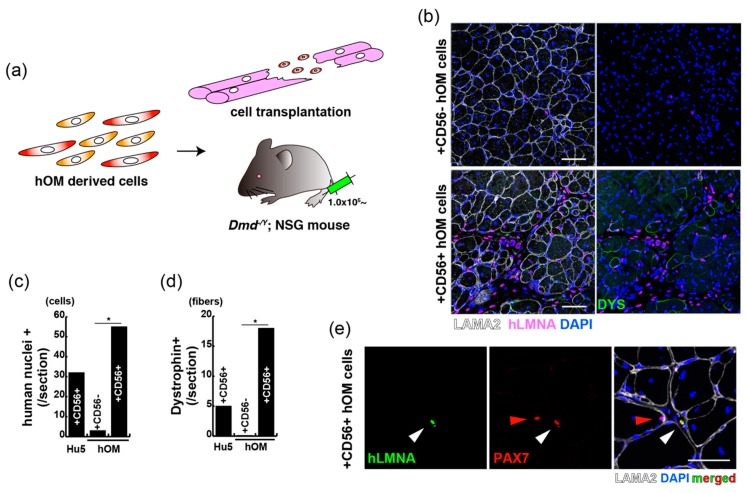
CD56-positive cells from orbicularis oculi muscle have regenerative capacity for Duchenne muscular dystrophy model mice. (**a**) A schematic model for the transplantation with sorted cells of orbicularis oculi muscle (hOM) into DMD-null (*Dmd^-/y^*) and NSG mice. (**b**) Immunostaining for LAMININ alpha-2 (LAMA2, labelled with Alexa647, white), human nuclear LAMIN A/C (hLMNA, labelled with Alexa594, pink), and DYSTROPHIN (DYS, labelled with Alexa488, green) in tibialis anterior (TA) muscles of *Dmd^-/y^*. NSG mice were injected with CD56-negative (upper panels) or CD56-positive (lower panels) cells derived from the orbicularis oculi 2 weeks after intramuscular engraftment. Total nuclei were stained with DAPI (blue). Scale bar, 50 μm. (**c**) The quantification of transplanted human nuclear cells (human nuclei+) in TA muscles engrafted with equal numbers of immortalized human myoblast Hu5 cells, and CD56-negative (+CD56−) or positive (+CD56+) cells from orbicularis oculi. *n* = 3 independent replicates; *p*-values are determined by *t*-test from a two-tailed distribution. * *p* < 0.01. Error bar indicates ±SEM. (**d**) The quantification of total DYSTROPHIN-positive regenerative myofibers on the section transplanted with equal numbers of indicated cells as in (**c**). *n* = 3 independent replicates; *p*-values are determined by *t*-test from a two-tailed distribution. * *p* < 0.01. Error bar indicates ±SEM. (**e**) Immunostaining for PAX7 (labelled with Alexa594, red), hLMNA (labelled with Alexa488, green), LAMA2 (labelled with Alexa647, white), and DAPI (blue) on a section of engrafted tibialis anterior with cultured CD56-positive cells from the orbicularis oculi. White arrow shows a transplanted PAX7+ cell. Red arrow shows intact mouse muscle satellite cell. Scale bar, 50 μm.

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
