# Peer review of "Human Skeletal Muscle Cells Derived from the Orbicularis Oculi Have Regenerative Capacity for Duchenne Muscular Dystrophy"

_ijms, 2019, doi:10.3390/ijms20143456_

Round 1
Reviewer 1 Report
The manuscript describes isolation of myogenic cells from eyelid surgeries using previously reported methods of selection. The authors demonstrate that myogenic activity is restricted to the CD56+ fraction and CD56+ CD82+ fraction, which is similar to what seen for other muscles.
A few points of clarification are suggested to make this manuscript stronger:
1) In figure 3D, CD56+ sorted cells were propagated and differentiated. How many passages typically can the CD56 positive cells be expanded with maintained myogenic activity and how many total cells can be typically generated? Does the karyotype change?
2) Figure 4. It seems that CD56+CD82+ cells need to be extracted immediately after surgery to select the myogenic cells from non myogenic cells. However in Figure 4C in hOM (aged p2) a population of CD56+ CD82- cells is present (4.6%). Is this population myogenic? What percentage of these cells are Pax7+?
3) In Figure 5 were CD56+ cells transplanted or CD56+CD82+ (which contain a higher percentage of Pax7 transcript). Please explain why.
4) In the methods "4.5 Grafting into dystrophin..." the authors should correct 20 ml of PBS to 20 microliters of PBS?
5) Figure S1. Please explain why there are higher cell numbers in 3% Oxygen but the percentage of CD56+CD82+ is higher in 20% oxygen. Please clarify what conditions would be better to expand myogenic cells and whether the transplanted CD56+ cells were grown in 20% or 3% oxygen or not expanded at all.
Author Response
The manuscript describes isolation of myogenic cells from eyelid surgeries using previously reported methods of selection. The authors demonstrate that myogenic activity is restricted to the CD56+ fraction and CD56+ CD82+ fraction, which is similar to what seen for other muscles.
A few points of clarification are suggested to make this manuscript stronger:
1) In figure 3D, CD56+ sorted cells were propagated and differentiated. How many passages typically can the CD56 positive cells be expanded with maintained myogenic activity and how many total cells can be typically generated? Does the karyotype change?
We have checked that CD56-positive cells from orbicular oculi of young patients have myogenic potential after 5 or 6 passages, however we could not detect any myogenic differentiated cells from CD56-positive cells derived from old patients after 2 or 3 passage (as not well-grown).
Primary cell culture with 1 sample of extra eyelid tissues were set on 4~5 dishes of 250px culture dish, then these dishes with expanded cells were analyzed with FACS. We normally gained
104~105 cells as CD56+ positives, and have not checked the karyotype, just examined Mycoplasma (all samples were negative).
2) Figure 4. It seems that CD56+CD82+ cells need to be extracted immediately after surgery to select the myogenic cells from non-myogenic cells. However in Figure 4C in hOM (aged p2) a population of CD56+CD82- cells is present (4.6%). Is this population myogenic? What percentage of these cells are Pax7+?
CD56+CD82- cells contains differentiated myogenic cells because they can be detected as MyHC positives. However we have not detected any PAX7+ cells in these cells by immunostaining, although Pax7 mRNAs were upregulated than CD56-CD82- double negative cells (Figure 3g).
3) In Figure 5 were CD56+ cells transplanted or CD56+CD82+ (which contain a higher percentage of Pax7 transcript). Please explain why.
We have transplanted CD56+, not CD56+CD82+ positive cells. Because we used immortalized myogenic Hu5 cells, which do not normally have PAX7 and are all CD56-positives, as positive control for Dystrophin recovery after cell transplantation. We mentioned these explanations in Results 2.4.
4) In the methods "4.5 Grafting into dystrophin..." the authors should correct 20 ml of PBS to 20 microliters of PBS?
5) Figure S1. Please explain why there are higher cell numbers in 3% Oxygen but the percentage of CD56+CD82+ is higher in 20% oxygen. Please clarify what conditions would be better to expand myogenic cells and whether the transplanted CD56+ cells were grown in 20% or 3% oxygen or not expanded at all.
It has been reported that low oxygen condition promotes the stemness of cultured mouse satellite cells (Liu et al., Development 2012). So we have tried to use the condition of low oxygen for cell culture, however we could not find this condition promoted high percentages of CD56+CD82+ cells (Figure S1d). We used CD56+ transplanted cells which were grown in normal 20% oxygen. We have mentioned those in Result 2.4.
Reviewer 2 Report
In this paper Yamanaka et al. propose orbicularis oculi muscle as a new source of human satellite cells. They clearly show that satellite cells are present in fibers from that muscle of human origin, that the cells can be cultured, and either sub-cultured, and that they maintain myogenic differentiation capacity. Importantly, when injected in vivo in TA muscle of mdx mice, dystrophin expression was restored and human cells, expressing Pax7, were found within the muscle, although in a non-canonical position.
The possibility to obtain human satellite cells from an easily accessible muscle, as from the extra eyelids, is of great interest for therapeutic approach, and this study, although very preliminary, appears to be promising.
As a general comment, I found sometime confusing the way the paper is presented. Moreover, the term “satellite cells” is sometimes used unproperly, and often it is not clear whether it refers to the bulk mononuclear cell population derived from the enzymatic digestion or to the CD56+ sorted cells.
In the introduction, many information is given as acquired, such as the choice of the molecular markers. People not familiar with the system, may not know which the biological features of CD56+ or CD82+, or both positive, cells, are; although references are given, just a little sentence explaining might be useful.
Figures 1 and 2 should be inverted: it is more reasonable to show that satellite cells are present in the candidate muscle first, and then describe how to isolate them.
Figure 3 is confusing: were CD56+ cells sorted out from a 7day bulk culture? If this is true, how they were detached, is not described. Moreover, will that mean that the few original satellite cells grew becoming the 8% of the cells, so that they could be sorted out? But in Figure 1 it was shown that the cells were expressing myogenin, so they were probably already differentiating. Moreover, the IF in Figure 3d, shows that many of the sorted cells differentiate in culture, as myosin-positive cells, however no sign of fusion is evident: is that because they do not fuse in culture? Again, in Figure 3f and g which are the cells shown: are those the freshly sorted cells from a previous culture? If this is the case, they must be shown before showing their differentiation ability.
Figure 4 also must be clearer described and discussed: if I understood correctly, subculturing human myogenic cells (bulk population? Sorted cells?) reduces significantly the percentage of CD56+ CD82+ cells, also depending on the age, is conceivable; however, given the very small yield of satellite cells from this muscle, it might be discussed how one can think to use these cells for therapeutic approach.
Figure 5 is actually the most important, showing that CD56+ cells can be engrafted into dystrophic muscle to restore dystrophin expression. Again, it is not clear which cell population was used: were the CD56+ cells sorted from the bulk culture after how many days in culture. Moreover, the unit of values in the ordinate axes is not given (are those absolute number, percentage respect to what?). The observation of the displaced position of Pax7+ cells is intriguing and might need further investigation. Would they be able to contribute to further repair, as being pure muscle stem cells? In a chronic disease as dystrophy, it would be difficult to assess, but it could be done in an acute model of muscle damage, to see whether in a repeated injury, human cells can contribute to repeated repair. Although it is not mandatory, it might be discussed.
The discussion should address each result more systemically, highlighting the implications of each.
Author Response
In this paper Yamanaka et al. propose orbicularis oculi muscle as a new source of human satellite cells. They clearly show that satellite cells are present in fibers from that muscle of human origin, that the cells can be cultured, and either sub-cultured, and that they maintain myogenic differentiation capacity. Importantly, when injected in vivo in TA muscle of mdx mice, dystrophin expression was restored and human cells, expressing Pax7, were found within the muscle, although in a non-canonical position. The possibility to obtain human satellite cells from an easily accessible muscle, as from the extra eyelids, is of great interest for therapeutic approach, and this study, although very preliminary, appears to be promising. As a general comment, I found sometime confusing the way the paper is presented. Moreover, the term “satellite cells” is sometimes used unproperly, and often it is not clear whether it refers to the bulk mononuclear cell population derived from the enzymatic digestion or to the CD56+ sorted cells.
In the introduction, many information is given as acquired, such as the choice of the molecular markers. People not familiar with the system, may not know which the biological features of CD56+ or CD82+, or both positive, cells, are; although references are given, just a little sentence explaining might be useful.
Following to reviewer’s suggestions, we have added the explanation for using CD56 or CD82 to isolate human myogenic cells in the introduction as below,
“It has been known that CD56 is the marker that enables to isolate human myogenic cells [5], and CD82 is novel marker for detecting human muscle stem cells [4,5].”
Figures 1 and 2 should be inverted: it is more reasonable to show that satellite cells are present in the candidate muscle first, and then describe how to isolate them.
We have changed the order of Figure1 and Figure2 as reviewer’s suggestion.
Figure 3 is confusing: were CD56+ cells sorted out from a 7day bulk culture? If this is true, how they were detached, is not described. Moreover, will that mean that the few original satellite cells grew becoming the 8% of the cells, so that they could be sorted out? But in Figure 1 it was shown that the cells were expressing myogenin, so they were probably already differentiating. Moreover, the IF in Figure 3d, shows that many of the sorted cells differentiate in culture, as myosin-positive cells, however no sign of fusion is evident: is that because they do not fuse in culture? Again, in Figure 3f and g which are the cells shown: are those the freshly sorted cells from a previous culture? If this is the case, they must be shown before showing their differentiation ability.
We have isolated CD56+ cells in Figure3a or CD56+CD82+ cells in Figure 3f and 3g from bulk culture for 7days. And then detached with Accutase (Nacalai). We have mentioned these procedures in Material and Methods.
These CD56+ sorted cells contain not only muscle stem cells but differentiating MYOGENIN+ myoblasts. We think that the CD56+CD82+ population, a half of CD56+ population in Figure 3f, mainly contains PAX7 positive cells.
We have changed pictures of Figure 3d because it was difficult to understand as reviewer’s suggestion. These cells were cultured to differentiate into Myosin+ cells as shown in new Figure 3d.
Figure 4 also must be clearer described and discussed: if I understood correctly, subculturing human myogenic cells (bulk population? Sorted cells?) reduces significantly the percentage of CD56+ CD82+ cells, also depending on the age, is conceivable; however, given the very small yield of satellite cells from this muscle, it might be discussed how one can think to use these cells for therapeutic approach.
In figure 4, as reviewer indicated, we have gained CD56+CD82+ cells as a few numbers of muscle stem cells from even grown conditions after primary culture. we have mentioned about those and further vision to use extra orbicular oculi muscles in the Discussion as below,
“These myogenic stem cells from orbicular oculi were not gained as massive scales from other muscle tissues, however we need not unnecessary surgical operation, and can plan with regularity, not accidentally, to have myogenic cells from orbicular oculi. Obtaining large quantity of muscle stem cells in culture would be next challenge”.
Figure 5 is actually the most important, showing that CD56+ cells can be engrafted into dystrophic muscle to restore dystrophin expression. Again, it is not clear which cell population was used: were the CD56+ cells sorted from the bulk culture after how many days in culture. Moreover, the unit of values in the ordinate axes is not given (are those absolute number, percentage respect to what?). The observation of the displaced position of Pax7+ cells is intriguing and might need further investigation. Would they be able to contribute to further repair, as being pure muscle stem cells? In a chronic disease as dystrophy, it would be difficult to assess, but it could be done in an acute model of muscle damage, to see whether in a repeated injury, human cells can contribute to repeated repair. Although it is not mandatory, it might be discussed.
We have used CD56-sorted cells from bulk population of extra eyelids which were cultured for 7 days. We have added these sentences in the Results 2.4.
Following to the reviewer’s advice, we have modified the graph in Figure 5c and 5d.
These transplanted cells might contribute into damaged myofibers again because we observed them on early stages of transplanted muscles after 2wks. We have discussed these further aspects after the transplantation in the Results 2.4.
The discussion should address each result more systemically, highlighting the implications of each.
Following to the reviewer’s suggestions, we have added quotes of each figure number in the Discussion.
Reviewer 3 Report
This manuscript written by Dr. Yamanaka et al has demonstrated the isolation and purification of myogenic cells from extra eyelid including orbicularis oculi muscles, and have defined the potential of these cells. The authors have also shown that these cells are the candidate source for cell transplantation therapy. This study has significance in a way that the results are from human derived cells, and the proposed technique, which can yield the high number of myogenic cells non-invasively, is highly demanded. Therefore, provided data were interesting.
However, the manuscript lacks the clarity in the data presentation. The authors need to describe the hypothesis, purpose, and the results of each experiments in a more logical and precise manner. In addition, the discussion over the obtained results is insufficient. The authors have not cited or mentioned the previous reports about the function of orbicularis oculi muscles and derived cells (McLoon, 1998; Sekulic-Jablanovic, 2016; Liu, 2018 etc.). It is necessary for the authors to cite these reports, and to discuss in further depth to clarify the significance of the current study. The reviewer has raised some concerns that need to be addressed and revised before the publication.
-The abbreviation “DMD” was used for anti-DYSTROPHIN in page 3 while it was also used for the abbreviation of Duchenne muscular dystrophy in page 6 (line 4) and page 7. The authors should revise the abbreviations of these words.
-In page 2 line 9, what is “non-selected myoblasts”?
-In page 2 line 10, the sentences “These data highlight a potential role…...in the maintenance of a pool of stem cells in vivo” were not based on the results of this study, since the engrafted cells did not form satellite cells as the authors have shown in Fig. 5e.
-In Fig. 2d, the nuclei in the myofiber should be stained with DAPI or Hoechst in order to confirm that Pax7 was expressed in the nuclei.
-In Fig. 2e, the authors have analyzed the expression of CD56 in eyelid muscle cells obtained from individuals with different ages, and have found a few numbers of CD56-positive cells. Did the authors culture the CD56-positive and -negative cells in vitro? Further, were CD56-positive cells purified myogenic cells? These experiments are quite interesting since there were few evidences that CD56-antibody could directly purify myogenic cells from the human tissue cells without the expansion in vitro.
-In Fig.3d, the authors should show the pictures of MF20 staining in CD56-negative cells as a comparison data.
-In Fig. 4, the purpose and the design of these experiments were obscure. The authors should describe more in detail.
-In Fig. 4a and 4c, the authors should discuss the reasons why the percentage of CD56-positive cells had decreased after several passages in vitro. Were these phenomena caused by the change in the proliferation ability of myogenic cells?
-In Fig. 4b and 4d, please show the interpretation why there was a decline of PAX7 transcript levels in CD56/CD82-positive cells after several passages. In addition, please discuss about the differences in PAX7 levels between young and aged cells.
-In page 5 line 19, please describe the reason why the cells were cultured under the hypoxic condition, and please cite the references. It is also needed to explain why the percentage of CD56/CD82-positive cells were decreased in hypoxic condition.
- In Fig. 5e, transplanted CD56-positive cells did not enter the typical satellite cell position, whereas the previous study by Liu et al have represented the formation of satellite cells and myofibers by engrafted cells (Tissue Eng Part C Methods, 2018). Please discuss the reason of the discrepancy.
-In Fig. 5e, high magnification images is needed since each cell was too small.
-In page 8 line 34, 20 ml is a typo. Please revise it.
Author Response
This manuscript written by Dr. Yamanaka et al has demonstrated the isolation and purification of myogenic cells from extra eyelid including orbicularis oculi muscles, and have defined the potential of these cells. The authors have also shown that these cells are the candidate source for cell transplantation therapy. This study has significance in a way that the results are from human derived cells, and the proposed technique, which can yield the high number of myogenic cells non-invasively, is highly demanded. Therefore, provided data were interesting.
However, the manuscript lacks the clarity in the data presentation. The authors need to describe the hypothesis, purpose, and the results of each experiments in a more logical and precise manner. In addition, the discussion over the obtained results is insufficient. The authors have not cited or mentioned the previous reports about the function of orbicularis oculi muscles and derived cells (McLoon, 1998; Sekulic-Jablanovic, 2016; Liu, 2018 etc.). It is necessary for the authors to cite these reports, and to discuss in further depth to clarify the significance of the current study. The reviewer has raised some concerns that need to be addressed and revised before the publication.
We thank this reviewer for detailed and valuable comments and advices. We hope that we could improve all points raised by the reviewer in this revised manuscript. We have added these papers as References and Discussion.
-The abbreviation “DMD” was used for anti-DYSTROPHIN in page 3 while it was also used for the abbreviation of Duchenne muscular dystrophy in page 6 (line 4) and page 7. The authors should revise the abbreviations of these words.
As reviewers’ suggestion, we have modified all abbreviations of DYSTROPHIN as DYS, and Duchenne muscular dystrophy as DMD.
-In page 2 line 9, what is “non-selected myoblasts”?
We have modified this sentence as “immortalized human myoblasts”.
-In page 2 line 10, the sentences “These data highlight a potential role…...in the maintenance of a pool of stem cells in vivo” were not based on the results of this study, since the engrafted cells did not form satellite cells as the authors have shown in Fig. 5e.
Following to the reviewer’s suggestion, we have modified the sentence of Introduction as follows, “highly value as a source of muscle stem cells in vivo”.
-In Fig. 2d, the nuclei in the myofiber should be stained with DAPI or Hoechst in order to confirm that Pax7 was expressed in the nuclei.
As reviewer’s suggestion, we have merged it to DAPI staining in new Figure 1.
-In Fig. 2e, the authors have analyzed the expression of CD56 in eyelid muscle cells obtained from individuals with different ages, and have found a few numbers of CD56-positive cells. Did the authors culture the CD56-positive and -negative cells in vitro? Further, were CD56-positive cells purified myogenic cells? These experiments are quite interesting since there were few evidences that CD56-antibody could directly purify myogenic cells from the human tissue cells without the expansion in vitro.
As shown in new Figure 1e, we have tried direct isolation of CD56-positive cells from extra eyelids again, however we have not succeeded to expand these sorted cells. CD56-positive cells were confirmed as FACS level, but there might be too small amount cells for expansion in vitro culture or be non-proliferating cells.
-In Fig.3d, the authors should show the pictures of MF20 staining in CD56-negative cells as a comparison data.
We have not checked that sorted cells as CD56-negative populations, were stained with MF20 in our handlings because it has been published that CD56-negative cells could not be detected as myogenic cells (Uezumi et al, 2016, Figure 1 in Stem Cell Reports). We have modified the picture of Figure 3d as clearer staining with MF20.
-In Fig. 4, the purpose and the design of these experiments were obscure. The authors should describe more in detail.
Following to the reviewer’s advice, we have added some description in the Results 2.3.
-In Fig. 4a and 4c, the authors should discuss the reasons why the percentage of CD56-positive cells had decreased after several passages in vitro. Were these phenomena caused by the change in the proliferation ability of myogenic cells?
In the case of mouse primary myoblasts, we have observed same situation that sorted myogenic cells are not well-proliferating after several passages because of the decline of the proliferation ability. We thought that human myogenic cells also have similar proliferative decline after passages.
-In Fig. 4b and 4d, please show the interpretation why there was a decline of PAX7 transcript levels in CD56/CD82-positive cells after several passages. In addition, please discuss about the differences in PAX7 levels between young and aged cells.
It seems like sorted CD56+CD82+ cells contain not only PAX7 positive cells but PAX7-negative myogenic cells. We understand that the expression level of PAX7 transcripts was declined by the attenuation of PAX7+ cells in CD56+CD82+ population of passaged and aged samples. We have mentioned these explanations in Results 2.4.
-In page 5 line 19, please describe the reason why the cells were cultured under the hypoxic condition, and please cite the references. It is also needed to explain why the percentage of CD56/CD82-positive cells were decreased in hypoxic condition.
It has been reported that low oxygen condition promotes the stemness of cultured mouse satellite cells (Liu et al., Development 2012). So we have tried to use the condition of low oxygen for cell culture, however we could not find this condition promoted high percentages of CD56+CD82+ cells in our handlings (Figure S1d). In our data, hypoxia condition promoted total numbers of cultured cells, however relative numbers of CD56+CD82+ cells did not show significant increase. We used CD56+ transplanted cells which were grown in normal 20% oxygen. We have mentioned those in Result 2.4 and this reference.
- In Fig. 5e, transplanted CD56-positive cells did not enter the typical satellite cell position, whereas the previous study by Liu et al have represented the formation of satellite cells and myofibers by engrafted cells (Tissue Eng Part C Methods, 2018). Please discuss the reason of the discrepancy.
Transplanted human myogenic cells might contribute into stem cell positions of myofibers later because we observed them on early stages of transplanted muscles after 2wks, just confirmed the recovery of DYSTROPHIN. We have discussed these further aspects after the transplantation in the Results 2.4.
-In Fig. 5e, high magnification images is needed since each cell was too small.
Following to the reviewer’s suggestion, we have modified these figures to magnified images.
-In page 8 line 34, 20 ml is a typo. Please revise it.
As the reviewer indicated, we have corrected it.
Round 2
Reviewer 3 Report
All comments and suggestions were precisely addressed, and some revisions and editorial corrections were incorporated in this new version.
I believe this manuscript is suitable for the publication.